# Prevalence and predictors of transfusion-transmitted infections among blood donor types at a teaching hospital in Ghana: Implications for haemovigilance

Alice Charwudzi[1]*, Edward Morkporkpor Adela[2,3], Kingsley Kwadwo Asare Pereko[4], Martin Ampofo[3], Aaron Fenuku[1], Daniel Edem Azumah[2], Abdul Raman Asemah[5], Edward Ahiakwah[2], Angela Animwaah Osei[2], Emmanuel Kobina Mesi Edzie[5]

1 Department of Haematology, School of Medical Sciences, University of Cape Coast, Cape Coast, Ghana, 2 Laboratory Department, Cape Coast Teaching Hospital, Cape Coast, Ghana, 3 Blood Bank, Cape Coast Teaching Hospital, Cape Coast, Ghana, 4 Department of Community Medicine, School of Medical Sciences, University of Cape Coast, Cape Coast, Ghana, 5 Department of Medical Imaging, School of Medical Sciences, University of Cape Coast, Cape Coast, Ghana

* a.charwudzi@uccsms.edu.gh

## Abstract

### Introduction

Transfusion-transmitted infections (TTIs), notably hepatitis B virus (HBV), hepatitis C virus (HCV), human immunodeficiency virus (HIV), and syphilis, remain a major threat to blood safety in resource-limited settings. Ghana mainly uses rapid diagnostic tests (RDTs) for initial screening, which may affect the accuracy of detecting TTIs.

### Objective

This study estimated the prevalence, identified sociodemographic and donor-type predictors of TTIs and compared the diagnostic yield of RDTs versus ELISA in a teaching hospital in the Central Region of Ghana.

### Methods

This retrospective study analysed 10,152 available blood donors' records screened from January 2022 to March 2024. Fixed-site donors were initially screened using RDTs (SD Biosensor Standard Q® HBsAg/HCV Ab, First Response® HIV 1-2.0, Advanced Quality™ Syphilis) followed by confirmatory ELISA testing for RDT-non-reactive samples. Mobile donors underwent ELISA (ChemWell® FUSION analyser) testing only. Multivariable logistic regression was used to identify independent TTI predictors.

**Data availability statement:** All relevant data are within the manuscript and its Supporting information files.

**Funding:** The author(s) received no specific funding for this work.

**Competing interests:** The authors have declared that no competing interests exist.

**Abbreviations:** TTIs, Transfusion-transmitted infections; RDs, Replacement donors; VDs, Voluntary donors; NAT, Nucleic acid testing; RDTs, Rapid Diagnostic Tests; SSA, Sub-Saharan Africa

## Results

The overall prevalence of TTIs (infection with at least one tested pathogen) was 16.5% (95% CI: 15.80-17.20; N = 1,675), with syphilis 8.4% (95% CI: 7.83-8.91; N = 850) being the most common. Voluntary donors had a lower TTI prevalence than replacement donors (10.6% vs 19.9%, p < 0.001). Repeat donors exhibited reduced risk of HBV (aOR: 0.254, 95% CI: 0.206-0.313, p < 0.001), HCV (aOR: 0.734, 95% CI: 0.568-0.949, p = 0.018), and syphilis (aOR: 0.486, 95% CI: 0.417-0.567, p < 0.001). However, donor type itself was not a significant predictor of TTIs after adjusting for sociodemographic variables. ELISA testing identified an additional 7.3% (95% CI: 6.67-8.01; N = 422/5,754) TTI cases among RDT non-reactive fixed-site donors (missed cases).

## Conclusion

The high prevalence of TTIs highlights persistent blood safety challenges. Repeat donation was independently protective, reducing risks of HBV, HCV, and syphilis. To improve blood safety, it will be essential to encourage regular voluntary donations. It will also require supplementing RDTs with ELISA where feasible, and strengthening haemovigilance systems, while accounting for the cost and logistical constraints. Although NAT is the gold standard for TTI detection, nationwide implementation in Ghana is currently not feasible.

## Introduction

Transfusion-transmitted infections (TTIs), including hepatitis B virus (HBV), hepatitis C virus (HCV), human immunodeficiency virus (HIV), and syphilis, remain major public health concerns globally, particularly in Sub-Saharan Africa (SSA), where prevalence rates are disproportionately high [1,2]. Despite global advances in blood safety, such as nucleic acid testing (NAT) and HBV vaccination, many low-and middle-income countries (LMICs), including Ghana, rely heavily on rapid diagnostic tests (RDTs) and replacement donation, increasing the risk of TTIs [2–4]. In high-income countries with stringent donor selection and advanced screening technologies such as NAT, the prevalence is typically below 1% [3–5]. In contrast, higher rates are consistently reported in SSA. In Ghana, overall TTI prevalence as high as 13–36% [6–9] has been documented, while comparable SSA settings report rates of 10–15% [1,10,11]. Ghanaian studies reported seroprevalence of HBV (3.1-13.2%), HCV (1.3-8.7%), HIV (1.6-4.5%), and syphilis (3.8-15.3%), indicating persistent gaps in blood safety [6–9]. Specifically, Walana et al., in their multiregional study (Oti, Greater Accra, Northern and Upper West Regions), reported HBV (6.6%), HCV (4.9%), HIV (2.9%), and syphilis (6.8%) [6]. In the Volta Region, Hadfield et al. reported HBV (3.1%), HCV (5.0.%), HIV (1.6%), and syphilis (6.4%) [7]. Another study by Addai-Mensah et al., in the Ashanti Region, found HBV (6.8%), HCV (1.3%), HIV (1.8%), and syphilis (3.8%) [8]. Lastly, Alomatu et al., in the Eastern Region,

found HBV (13.2%), HCV (8.7%), HIV (4.5%), and syphilis (15.3%) [9]. This disparity indicates the need to improve screening methodologies and blood donor management strategies in resource-constrained settings.

In Ghana, haemovigilance follows a structured yet evolving framework, with mandatory TTI screening primarily using serological assays [12]. The National Blood Service, Ghana (NBSG), governed by the National Blood Service Act, 2020 (Act 1042), oversees blood collection, testing, and distribution [13]. Although Ghana aims to transition to voluntary non-remunerated donations (VNRDs), replacement donation (wherein blood is donated by family or friends for specific patients) still constitutes 37–97% of donations in some regions [6,14]. However, replacement donations are often linked with higher TTI risk, as donors may conceal high-risk behaviours or engage in commercialised donation practices [1,15].

RDTs remain widely used because they are cost-effective and easy to use, but their limited sensitivity, especially during the window period, leads to missed early infections [16,17]. Enzyme-linked immunosorbent assay (ELISA) offers greater sensitivity, detecting an additional 5–6% of TTI cases missed by RDTs in Ghana and Nigeria [18,19]. NAT, the most sensitive method, can detect infections during the window period [20,21]. For example, in India, NAT identified an additional 0.2% of infections beyond those detected by ELISA (1.5% of overall TTI prevalence) [21]. Despite its superior performance, NAT remains underutilised in most SSA countries due to financial and logistical barriers [20].

Evidence on TTIs in Ghana is growing, but data gaps persist, particularly in the Central Region of Ghana, where donor-type disparities and RDT performance are not well characterised. This study estimates TTI prevalence, identifies sociodemographic and donor-type predictors, and compares the diagnostic yield of RDTs versus ELISA in routine screening at a teaching hospital in the Central Region of Ghana.

## Materials and methods

### Study design, site and period

This retrospective cross-sectional study analysed all potential and available blood donors' records from 1st January 2022–31st March 2024 at the Blood Bank Unit of Cape Coast Teaching Hospital (CCTH), Ghana. CCTH is the main tertiary referral centre in the Central Region of Ghana. The hospital's blood bank supports both fixed-site (in-hospital) and mobile (outreach) donation services. Key activities include testing of the major TTIs and cross-matching of safe blood. Donors' records were accessed from 1st September 2023–31st January 2025.

### Donor recruitment and classification

Donors were classified as:

1. Fixed-site donations: this was conducted at the hospital premises and included;

    (a) replacement/pre-deposit donors: individuals donating specifically to known patients (N = 6,450 (63.5%)) and

    (b) voluntary walk-in donors: community members who donated at the fixed-site (blood bank) without external prompting (N = 179 (1.8%)).

2. Voluntary mobile donors: individuals who participated in organised blood drives, mostly from secondary (boys-only, girls-only, and mixed-gender schools) and tertiary schools (N = 3,523 (34.7%)).

Donors were also categorised as first-time or repeat donors. Repeat donors were defined as individuals with at least one prior donation before the current study period.

### Donor information

Voluntary donors (VDs) comprised 179 (2.7%) voluntary walk-in donors at the fixed site and 3,523 (97.3%) voluntary mobile donors (outreach drives). These donors typically receive small refreshments, such as a cup of beverage and

biscuits to support post-donation care. Additionally, souvenirs such as customised pens, exercise books, tins of milk, and chocolate drinks are given to the donors from the blood bank or sponsors, consistent with findings by Asamoah-Akuoko et al. [22]. In contrast, replacement/pre-deposit donors, referred to as replacement donors in this study, did not receive incentives from the blood bank, as they donated for relatives or friends in need.

## Study population

**Eligibility criteria.** All potential blood donors with complete screening records (demographic details, donation history, TTI results, pre-donation RDT results for fixed-site donors and post-donation ELISA results) were included (N = 10,309). A total of 157 (1.5%) potential donors with incomplete data were excluded (Fig 1), leaving 10,152 for analysis.

## Testing protocols

**Fixed-site donors.**

1. Pre-donation RDT screenings used HBsAg (Standard Q), HCV Ab (Standard Q), HIV 1/2 Ab (First Response®), and syphilis Ab (Advanced Quality™ One-Step Anti-TP (TP/syphilis); N = 6,629 (64.3%).

2. Post-donation ELISA confirmation (Fortress Diagnostics 4th-generation assays) was performed for all RDT non-reactive samples (N = 5,754). According to the site protocol, RDT-reactive samples were not confirmed with ELISA. This reflects the national blood screening policy in Ghana, where all RDT-reactive donors are deferred immediately (permanently) to ensure maximum transfusion safety [23,24]. This lack of confirmatory testing may have overestimated prevalence due to possible false positives.

**Off-site (mobile donors).** Direct ELISA testing (no RDTs pre-screening) was performed post-donation, N = 3,523 (34.7%). Because fixed-site donors underwent RDT followed by ELISA (for negatives only), whereas mobile donors were screened by ELISA only, differences in diagnostic sensitivity could bias prevalence comparisons between donor groups.

ELISA testing was conducted using the ChemWell® FUSION automated analyser (Awareness Technologies, USA). Internal quality controls included daily calibration (inter-assay CV < 10%) and negative and positive controls in each assay run. The manufacturer's protocols were strictly followed. Kit specifications, including sensitivity and specificity, are detailed in S1 Table.

## Data collection and processing

Donor selection at CCTH followed the National Blood Service of Ghana guidelines [12]. This study's data team implemented pre- and post-entry validation and verification protocols to minimise entry errors. The principal investigator monitored the data collectors to ensure completeness. Trained research staff collected data using a standardised Google Sheet. Data was extracted from archived donor questionnaire sheets, laboratory registers, and crossmatch books using anonymised unique identification numbers assigned to eligible donors. While these unique identifiers facilitated data linkage for the ELISA and blood group subset, researchers did not retrieve any direct personal identifiers (including names or contact information) throughout the study. All data were fully de-identified before analysis.

Blood group data were available for 8,356 out of 10,152 blood donors (82.3% of the cohort), who were primarily TTI-negative donors whose donated blood was deemed safe for cross-matching. The blood groups were retrieved from the cross-matching records using their unique identification numbers.

Repeat donations were considered independent entries, as each donation was assigned a new identifier upon presentation. This approach aligns with the operational perspective of the blood bank, where each donation is screened independently to ensure safety. While this may overcount individuals who donated multiple times, it provides an accurate estimate of infection risk per donation, which is relevant to transfusion safety.

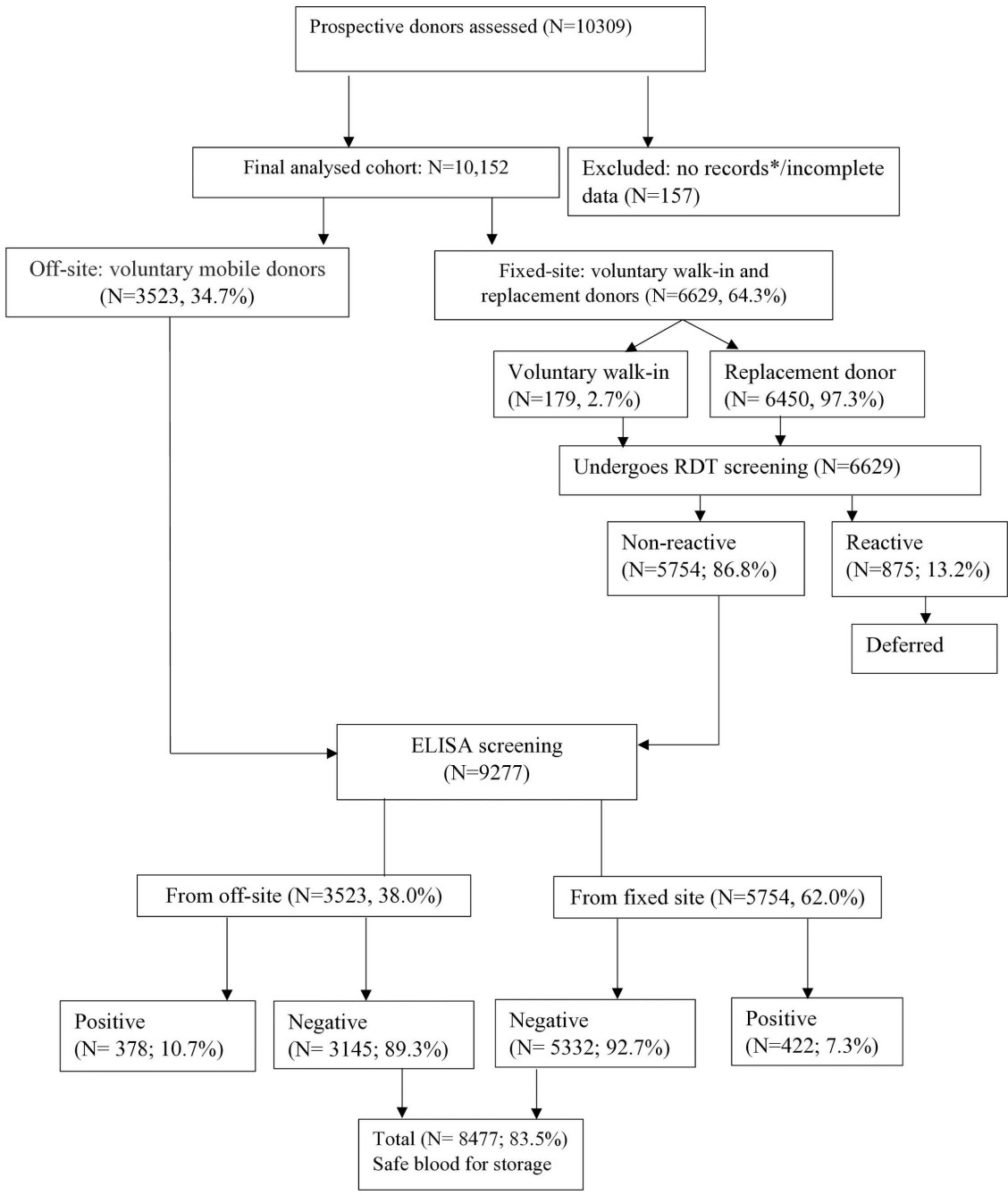

**Fig 1. Potential donors' recruitment and screening process.** Flowchart of the recruitment and screening process for 10,152 potential blood donors at a teaching hospital in the Central Region of Ghana. *Donors with incomplete data at initial extraction were excluded without record retention. Of 10,309 captured records, 157 (1.5%) with incomplete fields were removed, leaving 10,152 for analysis.

## Statistical analysis

SPSS version 29.0 was used for data analysis. Descriptive statistics were used to summarise categorical data as counts and percentages, and continuous variables as means ± standard deviation. Pearson's chi-square test was used to assess

associations between categorical variables. Binary and multivariable logistic regression models were used to identify predictors of TTIs, with results presented as adjusted odds ratios (aOR) and 95% confidence intervals (CI). Adjusted odds ratios were used as measures of association to estimate the likelihood of donors testing positive for any of the four TTIs, after adjusting for potential confounders. Covariates were age, gender, donation frequency, donor type and occupations, selected from the literature [10]. A high odds ratio indicated greater infection risk. All categorical covariates were entered as factors. A two-sided p-value < 0.05 was considered statistically significant. No formal correction for multiple testing was applied; therefore, results with p-values near the significance threshold should be interpreted cautiously. Graphics were generated using GraphPad Prism 10.2.3.

### Ethics approval

This study was approved by the Cape Coast Teaching Hospital (CCTH) Ethics Committee (REF: CCTHERC/EC/2023/174, dated 14th August 2023). An amendment to enable the data collection team to verify and finalise all missing or uncaptured data was approved on 30th January 2025. The study used pre-existing data, and participants were not directly engaged in any phase of the study; therefore, informed consent was not required. Anonymised data were analysed. All data were handled in compliance with Ghana's patient confidentiality regulations.

## Results

### Donor characteristics

The characteristics of the study population are shown in Table 1. A total of 10,152 potential blood donors with complete demographic records were included from January 2022 to March 2024. Of these, 6,450 (63.5%, 95% CI: 62.60-64.47) were replacement donors and 3,702 (36.5%, 95% CI: 35.52-37.40) were voluntary donors. Of the voluntary donors, 179 (4.8%, 95% CI: 4.20-5.60) were walk-in donors at the fixed site, while 3,523 (95.2%, 95% CI: 94.40-95.80) were voluntary mobile donors.

The median age was 26 years (Interquartile Range [IQR]: 20-32 years), and the male-to-female ratio was 4:1 (80.2% male, 19.8% female).

Voluntary donors were predominantly younger, with 2301 (62%, 95% CI: 60.58-63.70) aged 16-20. They were also more often female, 1488 (40.2%, 95% CI: 38.63-41.78) and mainly students, 3030 (82%, 95% CI: 80.57-83.06). In contrast, RDs were older, predominantly male 5927 (92%, 95% CI: 91.02-92.66), and largely employed 5188 (80%, 95% CI: 79.40-81.45). Voluntary donors were also more likely to be first-time donors, 2674 (72%, 95% CI: 70.77-73.65).

Similar demographic differences were observed by donation site: all mobile drives comprised VDs, whereas fixed-site collections (N = 6629) included both RDs, 6450 (97.3%, 95% CI: 96.91-97.69%) and a small proportion of voluntary walk-in donors, 179 (2.7%, 95% CI:2.31-3.09%) (S2 Table).

Of the total 2,011 female donors, 1,488 (74.0%, 95% CI: 72.08-75.91) were voluntary, and nearly half, 988 (49.1%, 95% CI: 46.94-51.31) were adolescents (16-20 years).

### Blood group distribution

Among the 10,152 potential blood donors, ABO and Rh D blood group data were available for 8,356 eligible donors, the majority of whom were screened and/or confirmed negative for TTIs by ELISA testing. Blood type O was the most prevalent, representing 5,146 (61.6%) donors. RhD-positive blood groups accounted for 91.5% of donors (Fig 2; S3 Table).

### TTI prevalence

The overall prevalence of TTIs, defined as seropositivity for at least one screened pathogen (HBV, HCV, HIV, or Syphilis), was 1,675 (16.5%, 95% CI: 15.78-17.22) (Table 1). Voluntary donors (N = 3,702) exhibited a lower overall TTI prevalence of 393 (10.6%, 95% CI: 9.62-11.61) compared to RDs (N = 6,450) who had a prevalence of 1,282 (19.9%, 95% CI: 18.90-20.85),

**Table 1. Relationship between sociodemographic characteristics and donor types among potential blood donors (N = 10,152).**

| | Voluntary donors N (%) | Replacement donors N (%) | Total, N (%) | P value |
|---|---|---|---|---|
| Total, N | N = 3702 | N = 6450 | N = 10152 | <0.001 |
| **Age group** | | | | <0.001 |
| 16 - 20 years | 2301 (62.2%) | 482 (7.5%) | 2783 (27.4%) | |
| 21 - 30 years | 1028 (27.8%) | 3210 (49.8%) | 4238 (41.7%) | |
| 31 - 40 years | 240 (6.5%) | 2049 (31.8%) | 2289 (22.5%) | |
| 41–60 years | 133 (3.6%) | 709 (11.0%) | 842 (8.3%) | |
| **Gender** | | | | <0.001 |
| Females | 1488 (40.2%) | 523 (8.1%) | 2011 (19.8%) | |
| Males | 2214 (59.8%) | 5927 (91.9%) | 8141 (80.2%) | |
| **D. History** | | | | <0.001 |
| First-time D. | 2674 (72.2%) | 2797(43.4%) | 5471 (53.9%) | |
| Repeat donors | 1028 (27.8%) | 3653 (56.6%) | 4681 (46.1%) | |
| **Occupation** | | | | <0.001 |
| Student | 3030 (81.8%) | 754 (11.7%) | 3784 (37.3%) | |
| Worker | 639 (17.3%) | 5188 (80.4%) | 5827 (57.4%) | |
| Unemployed | 13 (0.4%) | 300 (4.7%) | 313(3.1%) | |
| Others * | 20 (0.5%) | 208 (3.2%) | 228 (2.2%) | |
| **Overall TTI[#]** | | | | <0.001 |
| Positive | 393 (10.6%) | 1282 (19.9%) | 1675 (16.5%) | |
| Negative | 3309 (89.4%) | 5164 (80.1%) | 8473 (83.5%) | |

The sociodemographic characteristics of potential blood donors, stratified by donor type (voluntary vs. replacement). % = percentage.

*Others include apprentices, retirees, prisoners, and individuals who did not specify their occupation. D. = Donors or Donation, TTI = Transfusion-transmitted infections.

[#]Overall TTIs positivity was based on donors reactive/positive for at least one infectious marker.

p<0.001. Of the TTIs screened in this study, syphilis was the most common infection in both RDs and VDs, with a prevalence of 850 (8.4%, 95% CI: 7.84-8.90). The details of the different types of TTIs and their distribution among the two donor types are shown in Fig 3; S3 Table. Furthermore, compared with voluntary mobile donors, odds of TTI seropositivity did not differ significantly among voluntary walk-in donors (COR: 0.76, 95% CI 0.44-1.31, p = 0.32), but were significantly higher among RDs (COR: 2.06, 95% CI 1.83-2.33, p<0.001).

## Co-infections

Among the 10,152 potential blood donors, 149 (1.5%, 95% CI: 1.23-1.70) tested positive for two or more TTIs (Fig 4). Each co-infected donor was counted only once in this analysis (Fig 4). Rates did not differ significantly between voluntary (N = 3,702), with 57 (1.5%; 95% CI: 1.18-1.99), and replacement donors (N = 6,450), with 92 (1.4%; 95% CI: 1.15-1.75) (p = 0.648). The odds of co-infection were similar between voluntary and replacement donors (OR: 0.93, 95% CI: 0.66-1.29).

The most frequent co-infection combination was HBV and syphilis in (N = 79, 53.0% of all co-infections), occurring in 0.8% of all donors (95% CI: 0.61-0.95), Fig 4.

## Predictors of TTI positivity

Univariate analysis revealed significant associations between donor characteristics and the risk of infection (Table 2). Replacement donors were about twice as likely to have HBV (COR: 2.08; 95% CI: 1.68-2.58) and syphilis (COR: 2.34; 95% CI: 1.97-2.77) compared to VDs. Male donors showed elevated risks for HBV (COR: 1.76; 95% CI: 1.35-2.30) and

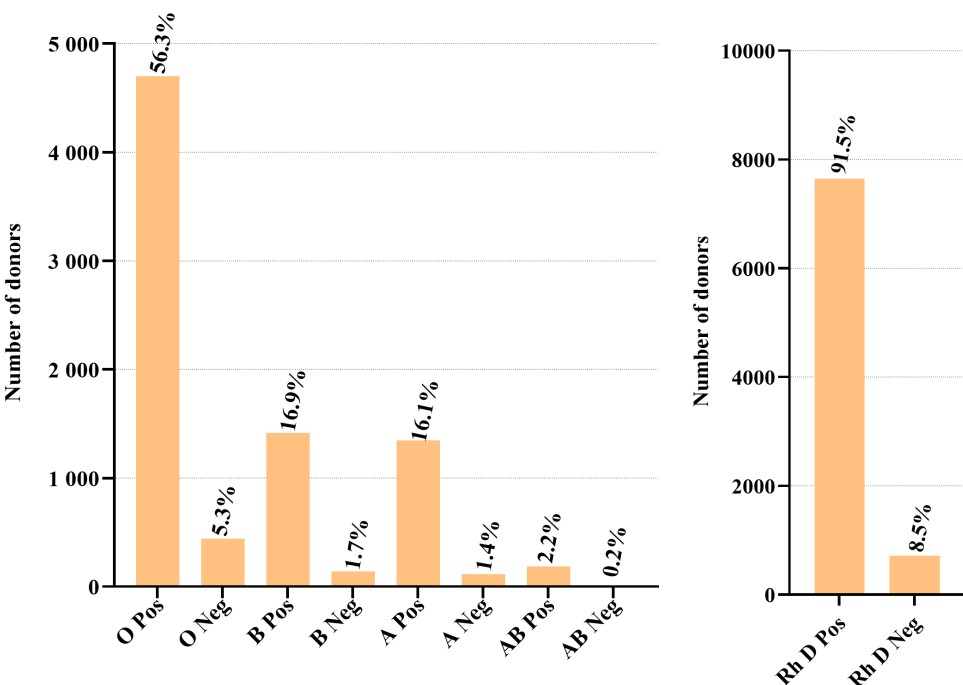

**Fig 2. Blood donors' ABO and RhD blood groups distribution for 8,356 donors.**

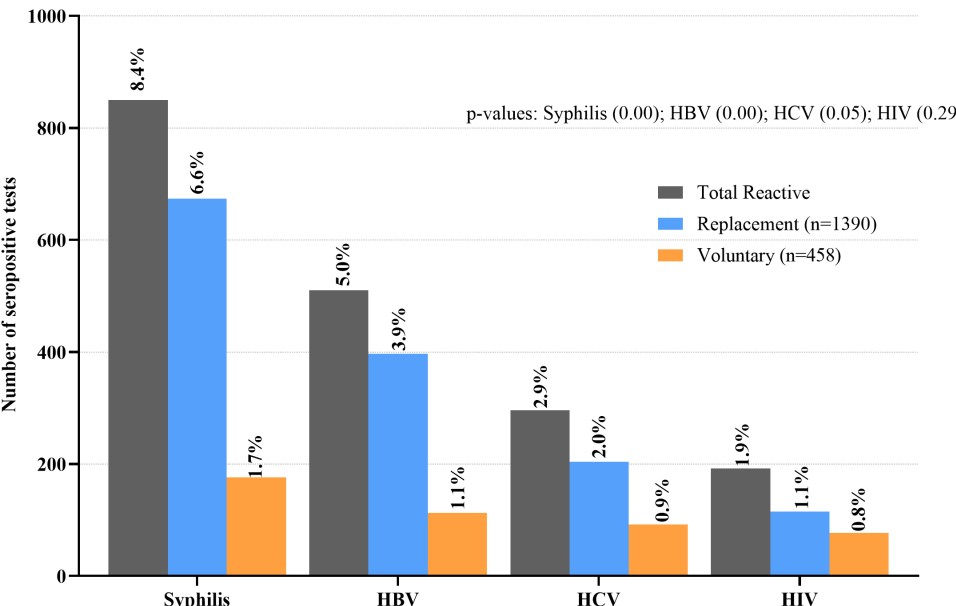

**Fig 3. The seroprevalence rates of syphilis, HBV, HCV, and HIV among replacement and voluntary donors, based on the number of reactive/ positive tests with RDTs and ELISA results. NB:** Infection-specific rates include co-infected donors in each TTI category. Total number of donors, N = 10,152 (Replacement, N = 6,450; Voluntary, N = 3,702).

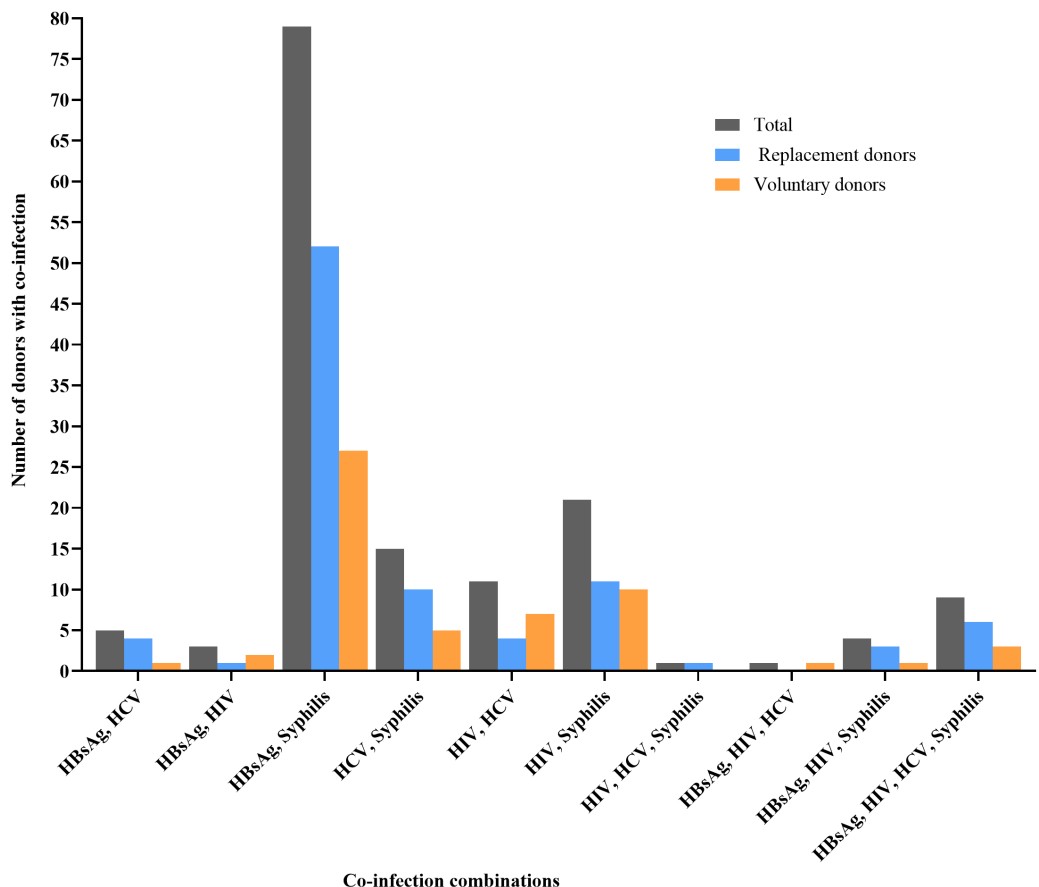

**Fig 4. Comparing co-infection rates among replacement and voluntary blood donors.** The total number of infection-specific positives (1,848) was greater than the number of individual TTI-positive donors (1,675), indicating that some donors were infected with more than one pathogen.

syphilis (COR: 2.99; 95% CI: 2.32-3.84) compared to females. Repeat donors were less likely to test positive for HBV, HCV, and syphilis than first-time donors.

Multivariable analysis (Table 3) showed that donors aged 41-60 years had the highest risk of HBV (aOR: 2.19; 95% CI: 1.42-3.39) and syphilis (aOR: 3.50; 95% CI: 2.48-4.95). Male donors were also more likely to have HBV (aOR: 1.35; 95% CI: 1.01-1.80) and syphilis (aOR: 2.11; 95% CI: 1.62-2.76) than females. Repeat donors had significantly lower infection risks compared to first-time donors: 75% lower odds for HBV (aOR: 0.25; 95% CI: 0.21-0.31), 51% lower for syphilis (aOR: 0.49; 95% CI: 0.42-0.57), and 23% lower for HCV (aOR: 0.77; 95% CI: 0.61-0.97; p<0.001). Donors who were employed (aOR: 2.76; 95% CI: 1.92-3.98) or in other occupations (aOR: 3.43; 95% CI: 1.88-6.23) had higher odds of HBV than students. Although crude analysis indicated a higher prevalence among RDs, donor type was not an independent predictor after adjusting for age, gender, donation frequency, donor type and occupation.

HIV showed no significant associations with demographic variables, as presented in Table 3. Donors with three or more previous donations had the lowest risk of infection (COR: 0.55; 95% CI: 0.45-0.67) (Table 4).

## Performance for screening methods

Comparing RDT and ELISA tests revealed significant differences in detection rates among fixed-site donors, which included both replacement and voluntary walk-in donors (N=6,629). Initial pre-donation RDTs screening identified 875 TTI-positive cases (13.2%; 95% CI: 12.39-14.02).

**Table 2. Prevalence of transfusion-transmitted infections (TTIs) among potential blood donors stratified by sociodemographic characteristics (N = 10,152).**

| Donor Characteristics | HBV | | HCV | | HIV | | Syphilis | | Total |
|---|---|---|---|---|---|---|---|---|---|
| (N=10152) | N (%) | COR (95%CI) | N (%) | COR (95%CI) | N (%) | COR (95%CI) | N (%) | COR (95%CI) | N (%) |
| **Age-group (years)** | | | | | | | | | |
| 16-20 (N=2783) | 78 (0.8) | Ref | 80 (0.8) | Ref | 61 (0.6) | Ref | 112 (1.1) | Ref | 331 (3.3) |
| 21-30 (N=4238) | 216 (2.1) | 1.86 (1.43-2.42) | 143 (1.4) | 1.18 (0.89-1.56) | 73 (0.7) | 0.78 (0.56-1.10) | 311 (3.1) | 1.89 (1.51-2.36) | 743 (7.3) |
| 31-40 (N=2289) | 152 (1.5) | 2.47 (1.87-3.26) | 51 (0.5) | 0.77 (0.54-1.10) | 44 (0.4) | 0.88 (0.59-1.29) | 286 (2.8) | 3.41 (2.72-4.27) | 533 (5.3) |
| 41-60 (N=842) | 64 (0.6) | 2.85 (2.03-4.01) | 22 (0.2) | 0.91 (0.56-1.46) | 14 (0.1) | 0.75 (0.42-1.36) | 141 (1.4) | 4.80 (3.69-6.23) | 241 (2.4) |
| **Gender** | | | | | | | | | |
| Female (N=2011) | 64 (0.6) | Ref | 53 (0.5) | Ref | 46 (0.5) | Ref | 69 (0.7) | Ref | 232 (2.3) |
| Male (N=8141) | 446 (4.4) | 1.76 (1.35-2.30) | 243 (2.4) | 1.14 (0.84-1.54) | 146 (1.6) | 0.78 (0.56-1.09) | 781 (7.7) | 2.99 (2.32-3.84) | 1616 (15.9) |
| **Donation History** | | | | | | | | | |
| First (N=5471) | 372 (3.7) | Ref | 178 (1.8) | Ref | 105 (1.0) | Ref | 493 (4.9) | Ref | 1148 (11.3) |
| Repeat (N=4681) | 138 (1.4) | 0.42 (0.34-0.51) | 118 (1.2) | 0.77 (0.61-0.97) | 87 (0.9) | 0.97 (0.73-1.29) | 357 (3.5) | 0.83 (0.72-0.96) | 700 (6.9) |
| **Donor Types** | | | | | | | | | |
| VDs (N=3702) | 113 (1.1) | Ref | 92 (0.9) | Ref | 77 (0.8) | Ref | 176 (1.7) | Ref | 458 (4.5) |
| RDs (N=6450) | 397 (3.9) | 2.08 (1.68-2.58) | 204 (2.0) | 1.28 (1.00-1.65) | 115 (1.1) | 0.86 (0.64-1.144) | 674 (6.6) | 2.34 (1.97-2.77) | 1390 (13.7) |
| **Occupation** | | | | | | | | | |
| Student (N=3784) | 95 (0.9) | Ref | 101 (1.0) | Ref | 77 (0.8) | Ref | 154 (1.5) | Ref | 427 (4.2) |
| Worker (N=5827) | 385 (3.8) | 2.75 (2.19-3.45) | 179 (1.8) | 1.16 (0.90-1.48) | 102 (1.0) | 0.86 (0.64-1.16) | 655 (6.5) | 2.99 (2.49-3.58) | 1321 (13.0) |
| Unemployed (N=313) | 13 (0.1) | 1.68 (0.93-3.04) | 8 (0.1) | 0.96 (0.46-1.98) | 9 (0.1) | 1.43 (0.71-2.87) | 26 (0.3) | 2.14 (1.39-3.29) | 56 (0.6) |
| **Others* (N=228)** | 17 (0.2) | 3.13 (1.83-5.34) | 8 (0.1) | 1.33 (0.64-2.76) | 4 (0.0) | 0.86 (0.31-2.37) | 15 (0.1%) | 1.66 (0.96-2.87) | 44 (0.4) |

The prevalence of HBV, HCV, HIV, and syphilis among blood donors is categorised by age, gender, donation history, donor type, and occupation.
VDs = Voluntary donors, RDs = replacement donors, Ref = Reference variable; COR = Crude Odds Ratio.

Subsequent ELISA testing of 5,754 RDT non-reactive donors identified an additional 422 TTI-positive cases (7.3%, 95% CI: 6.66-8.01). These were false negatives that RDT alone would have missed.

Among these ELISA-only positive cases, syphilis was the most common infection, accounting for 232 cases (4.0%, 95% CI: 3.52-4.54) of the 5,754 RDT-negative donors tested. Furthermore, 23 donors (0.4%, 95% CI: 0.24-0.56) were found to have co-infections missed by RDTs.

Overall, supplemental ELISA testing led to a 48.2% increase in case detection (422/875 additional cases).

**Table 3. Multivariable predictors of transfusion-transmitted infections (TTIs) among potential blood donors (N = 10,152).**

| N = 10152 | HbsAg | | HCV | | HIV | | Syphilis | |
|---|---|---|---|---|---|---|---|---|
| | P-val | aOR (95% CI) | P-val | aOR (95% CI) | P-val | aOR (95% CI) | P-val | aOR (95% CI) |
| **Age-group (years)** | | | | | | | | |
| 16-20 (N = 2783) | Ref | | | | | | | |
| 21-30 (N = 4238) | **0.045** | 1.420 (1.008-2.000) | 0.690 | 0.928 (0.643-1.340) | 0.323 | 0.798 (0.510-1.248) | **0.008** | 1.467 (1.106-1.947) |
| 31-40 (N = 2289) | **0.003** | 1.791 (1.215-2.638) | **0.018** | 0.564 (0.351-0.907) | 0.776 | 0.921 (0.523-1.624) | **<0.001** | 2.432 (1.771-3.341) |
| 41-60 (N = 842) | **<0.001** | 2.193 (1.417-3.392) | 0.188 | 0.680 (0.383-1.208) | 0.537 | 0.797 (0.388-1.638) | **<0.001** | 3.504 (2.479-4.953) |
| **Gender** | | | | | | | | |
| Female (N = 2011) | Ref | | | | | | | |
| Male (N = 8141) | **0.042** | 1.350 (1.011-1.803) | 0.668 | 1.075 (0.773-1.496) | 0.239 | 0.798 (0.549-1.161) | **<0.001** | 2.110 (1.615-2.756) |
| **Donation patterns** | | | | | | | | |
| First-time (N = 5471) | Ref | | | | | | | |
| Repeat (N = 4681) | **<0.001** | 0.254 (0.206-0.313) | **0.018** | 0.734 (0.568-0.949) | 0.671 | 1.073 (0.775-1.484) | **<0.001** | 0.486 (0.417-0.567) |
| **Donor type** | | | | | | | | |
| VDs (N = 3702) | Ref | | | | | | | |
| RDs (N = 6450) | 0.907 | 0.982 (0.726-1.328) | 0.097 | 1.372 (0.944-1.994) | 0.784 | 0.940 (0.605-1.461) | 0.389 | 1.108 (0.878-1.397) |
| **Occupation** | | | | | | | | |
| Student (N = 3784) | Ref | | | | | | | |
| Worker (N = 5827) | **<0.001** | 2.760 (1.915-3.979) | 0.203 | 1.293 (0.870-1.920) | 0.913 | 1.029 (0.619-1.709) | **<0.001** | 1.704 (1.287-2.255) |
| Unemployed (N = 313) | 0.119 | 1.677 (0.875-3.212) | 0.766 | 0.888 (0.406-1.941) | 0.178 | 1.723 (0.781-3.802) | 0.135 | 1.437 (0.893-2.312) |
| Others* (N = 228) | **<0.001** | 3.426 (1.884-6.232) | 0.458 | 1.346 (0.614-2.951) | 0.978 | 1.015 (0.345-2.988) | 0.769 | 1.091 (0.609-1.956) |

Multivariable logistic regression analysis identifying predictors of hepatitis B virus (HBV), hepatitis C virus (HCV), human immunodeficiency virus (HIV) and syphilis among potential blood donors. Ref = Reference, aOR = adjusted odds ratios, VDs = voluntary donors; RDs = replacement donors. Others include apprentices, retirees, prisoners, and individuals who did not specify their occupation.

**Table 4. Donation frequency as a predictor of transfusion-transmitted infections (TTIs) prevalence among repeat blood donors (N = 4,451).**

| Predictor | Prev N (%) | Univariable analysis | |
|---|---|---|---|
| | | p-value | (Exp(B) OR (95% CI) |
| **Donation frequency (N = 4451)** | | | |
| 1 donation (N = 1702) | 300 (6.8%) | | 1 (Reference) |
| 2 donations (N = 1056) | 136 (3.1%) | **<0.001** | 0.685 (0.551- 0.853) |
| >2 donations (N = 1693) | 180 (4.0%) | **<0.001** | 0.552 (0.452-0.673) |

Univariable logistic regression analysis was performed on 4,451/4,681 (95.1%) of all repeat potential blood donors with available data on the number of previous donations. OR = Odds ratio, CI = Confidence Interval, Prev = prevalence; Exp(B) = exponentiated coefficient.

## Discussion

Blood transfusion remains a critical life-saving intervention, yet it carries the inherent risk of transmitting infectious agents, particularly in resource-limited settings where screening infrastructure is suboptimal [2,17]. This study investigated the prevalence and predictors of TTIs among 10,152 potential blood donors at a tertiary hospital in the Central Region of Ghana. It also compared voluntary donors (VDs) and replacement donors (RDs) while evaluating the performance of RDTs against ELISA. The findings reveal three critical public health challenges: persistently high TTI prevalence, demographic disparities in infection risk, and under-detection by RDTs.

## The burden of TTIs and donor type disparities

The overall TTI prevalence, defined as positivity for at least one of the four screened pathogens (HBV, HCV, HIV, or syphilis), was 16.5% which is lower than the 36.2% reported in the Eastern region of Ghana [9]. However, the current rate aligns with the 13.5 - 21.0% previously reported in other regions of Ghana [6–8], and 15.0% in Nigeria [1], but remains higher than 10.1% in Tanzania [10]. In contrast, the World Health Organisation (WHO) reports <1% in high-income countries [4,5]. These disparities point to weaknesses in blood safety systems and reflect broader differences in healthcare infrastructure, screening, and prevention efforts [4,5].

Replacement donors in this study exhibited relatively higher infection rates than voluntary donors (19.9% vs. 10.6%, p < 0.001), consistent with findings from Ghana (23.5% vs. 3.5%) [8], Tanzania (12.3% vs. 9.5%) [10], and Pakistan (5.4% vs. 3.9%) [8,10,15]. This reinforces evidence that replacement donation models are associated with increased TTI risk.

Higher TTI rates among RDs may partly reflect commercial donors disguising themselves as RDs. Unlike voluntary donors, RDs often donate under pressure, which can encourage the concealment of high-risk behaviours [1,22]. In Nigeria, commercial donors have shown TTI prevalence as high as 62.9% [1]. In this study, the predominance of repeat donors among RDs (56.6%) and high unemployment (Table 1) mirror patterns seen among commercial donors, who frequently donate for income [1].

## The prevalence of syphilis, HBV, HCV and HIV

In this present study, syphilis was the most common infection (8.4%), Fig 3, higher than 3.8% [8], 6.4% [7], and 6.8% [6] in previous comparable studies in Ghana, 3.1% in Nigeria [1] and 1.9% in Tanzania [10]. Population-based syphilis estimates of five SSA countries reported ≤3% [25].

The differences across studies warrant careful consideration because they may reflect variations in testing methodology. In this current study, 64.3% of blood samples were screened using both RDTs and ELISA (Fig 1), whereas other studies relied on only one method (RDTs only or ELISA only) [1,6,8,10]. The assays used in this study detect treponemal antibodies, which persist for life even after successful treatment, potentially inflating prevalence. At the study centre, any TTI-positive result leads to permanent donor deferral; therefore, non-treponemal reagents are not routinely used for screening or confirmation. Also, the donor screening policy does not allow confirmation of an RDT-reactive sample by ELISA [23,24]. While these approaches prioritise blood safety, they may contribute to the overestimation of syphilis prevalence.

RDTs also show variable specificity, resulting in false positives [17,26]. A study in Ghana found that only 31.7% of RDT-reactive samples were confirmed positive using more specific assays (non-treponemal tests and enzyme immunoassay) [26]. This may partly explain the high prevalence of syphilis (Fig 1). Broader epidemiological factors, including higher rates of sexually transmitted infections (STIs) among certain male-dominated occupational groups, whose members more often donate as RDs, may also contribute to the higher rate among RDs [27,28].

The higher syphilis prevalence of 15.3% by Alomatu et al in Ghana may partly be attributed to their extremely small sample size [9].

HBV (5.0%) and HCV (2.9%) were the second and third most common TTIs. Walana et al. reported a similar but slightly higher prevalence of HBV (6.6%) and HCV (4.9%) in Ghana [6]. Systematic reviews conducted in Ghana estimated the national HBV prevalence of 8.4% [29] and HCV at 3.0% [30], higher than in this current study, confirming the endemic nature of these viruses in Ghana. In contrast, developed countries report much lower prevalence, e.g., HBV at 0.6% [31] and HCV at 0.9% [32], in Spain.

HIV prevalence was the lowest (1.9%) and consistent across donor categories (VDs, 0.8% vs RDs, 1.1%; p = 0.290) in this study. This result aligns with national averages (1.5%) [33] and other similar Ghanaian and SSA reports among blood donors (1.8-3.5%). Similar HIV prevalence across donor types may reflect national prevention efforts and a stabilised

epidemic [7,8,10,34]. Although not statistically significant, the elevated risk of HIV among female donors and adolescents (aged 16-20 years) (Tables 2 and 3) is consistent with UNAIDS reports [33].

## Predictors of TTI positivity

In the unadjusted analysis, RDs had significantly higher odds of HBV and syphilis compared to VDs (COR: 2.08 and 2.34), respectively (Table 2). However, after adjusting for age, gender, occupation, and donation history, donor type was no longer statistically significant (Table 3). This suggests that donor type itself was not an independent predictor, but risk was more strongly influenced by other sociodemographic factors like age, sex, donation frequency and occupation [27].

Differences in prevalence may also reflect demographic variation by donation site: mobile donors were younger, more often female, and predominantly students, while fixed-site donors were older, predominantly male, and employed.

The strongest protective factor was repeat donation. Compared to first-time donors, repeat donors had lower adjusted odds of HBV (aOR: 0.25), HCV (aOR: 0.73), and syphilis (aOR: 0.49) (Table 3), all with p-values less than 0.05. Repeat donation was protective across donor categories, despite the predominance of repeat donors among RDs (56.6%). This trend matches findings from Spain, where first-time donors had higher adjusted odds of 19 (95% CI: 11–35) [35]. Repeat donors are more familiar with donor eligibility criteria and are more likely to disclose risk behaviours accurately [11,36]. Over time, high-risk individuals may be deferred or self-exclude, resulting in a progressively safer donor population [8].

Other independent predictors included gender and occupation. Male donors had significantly higher adjusted odds of HBV and syphilis, partly consistent with earlier studies in Ghana and other SSA countries [7,10]. Student donors had lower odds of HBV and syphilis compared to employed and unemployed individuals.

Most VDs in this study were young students; the majority were recruited through school-based donation drives, yielding a lower-risk donor pool, partly reflecting trends reported in Ghana [6,8] and Pakistan [15]. Expanding outreach to young non-students may further support a safer donor base.

## Screening methods performance

The current study highlights the limitations of relying solely on RDTs for TTI screening. ELISA identified an additional 422 TTI-positive cases (7.3%) missed by RDTs, aligning with earlier findings from Ghana (6.2% discrepancy) [19] and Nigeria (4.9%) [18].

Although RDTs are cost-effective and simple to use, they often lack the sensitivity to detect early-stage infections and window period cases. This was most evident for syphilis, the most frequently missed infection in this study, reinforcing ELISA's superiority [16,17,19].

The implementation of ELISA nationwide is feasible in Ghana, though cost remains the main barrier [20]. The total setup cost is estimated locally at ≈ GHS 210,000-420,000 ($17,100-34,200) per centre, covering an automated ELISA analyser, cold-chain storage including 2-8 °C and −20°C, and infrastructure upgrades [37]. Training each technical staff member adds ≈ GHS 3,000-5,000 ($240-410). Recurrent costs are ≈ GHS 32-60 ($2.50-5.0) per test, comprising reagents, controls, and consumables [2,21,38]. NAT, though more sensitive and able to further shorten the window period [21], requires significantly higher investment. Automated platforms cost over GHS 250,000-600,000 ($20,400-48,900), with additional expenses for reagents, infrastructure, and training, making widespread adoption less feasible in the short term [20,39]. These local estimates are consistent with published reports highlighting cost and logistical barriers to NAT and ELISA adoption in low-resource settings [2,20]. These financial and technical barriers necessitate phased adoption to ensure sustainable integration into national screening programs.

## Co-infections

Co-infections, though less frequent, are still clinically relevant. Among all donors, 1.5% (Fig 4) tested positive for two or more TTIs, comparable with 2.3% reported in the Eastern region of Ghana [9], but higher than 0.1% in Eritrea [40]. Rates did not significantly differ between RDs and VDs (Fig 4). The most frequent combination was HBV and syphilis. This

reflects overlapping transmission routes, donor selection practices, general population prevalence, and assay sensitivity [19,40]. These patterns suggest the need for donor education and multi-pathogen screening.

**Limitations**

1. The retrospective and single-centre design may limit the generalisability and introduce bias due to incomplete or inaccurate records. Manual data extraction may also have introduced transcription errors.

2. Use of RDTs for initial screening at fixed sites may have inflated TTI prevalence, as RDT-reactive samples were not ELISA-confirmed, and RDTs generally have lower specificity than ELISA.

3. Diagnostic differences between fixed-site (RDT and ELISA) and mobile (ELISA-only) strategies may have biased prevalence comparisons. The diagnostic accuracy remains incomplete as variation in test sensitivity and specificity between RDTs and ELISA could have led to differential detection rates. Therefore, the comparisons between fixed-site and mobile strategies should be interpreted with caution.

4. Repeat donations were considered independent entries, as each donation is assigned a unique identifier upon presentation, limiting donor-level tracking.

5. No formal correction was applied for multiple comparisons, as analyses were based on pre-specified variables informed by existing literature. However, multiple comparisons may increase the risk of false positives, and results near significance should be interpreted with caution.

Future studies should consider multi-centre designs and explore the cost-effectiveness of integrating NAT into routine screening protocols.

**Policy considerations and practical recommendations**

1. Enhance recruitment of low-risk populations, especially repeat donors, and students through outreach and education. Improvement in non-financial incentives to promote regular voluntary donations.

2. Adopt dual testing (RDT and ELISA) for all donors, with phased integration of NAT as resources allow.

3. Promote collaboration among health authorities, educational institutions, non-governmental organisations (NGOs), and the private sector to secure sustained support and resources for a safer blood supply.

## Conclusion

TTI prevalence remains high among blood donors in Ghana, although repeat donors exhibit significantly lower infection risks. Voluntary donors had lower TTI rates than replacement donors; however, donor type was not an independent predictor of TTI positivity after adjusting for sociodemographic factors. The adoption of ELISA enhanced detection compared with RDTs. To strengthen blood safety, it is imperative to encourage regular voluntary donations, adopt more sensitive screening methods like ELISA, and reinforce haemovigilance systems nationwide. Although NAT remains the gold standard for TTI detection, its nationwide implementation in Ghana is currently not feasible due to logistical and cost constraints.

## Supporting information

**S1 Table. Specifications and performance metrics of screening kits used for the detection of transfusion-transmitted infections (TTIs).**
(DOCX)

**S2 Table. Relationship between sociodemographic characteristics and donor types among potential blood donors.**
(DOCX)

**S3 Table. Supporting data.**
(XLSX)

## Acknowledgments

We sincerely appreciate the staff of the Laboratory and Blood Bank Unit at Cape Coast Teaching Hospital (CCTH) for their invaluable technical support during data collection. In particular, we extend our gratitude to Dr Theophilus Mensah Ofori, Ms Ernestina Arthur, and Mr Ignitius Attah-Kwoffie for their contributions.

We are also deeply grateful to Mr Bismark Darko of the Haematology Department, School of Medical Sciences, University of Cape Coast, for his assistance with data collection and management. Our thanks also go to Mr Ebenezer Pappoe of the Microbiology Department, School of Medical Sciences, University of Cape Coast, for his support in data collection and management.

Additionally, we acknowledge the laboratory unit of the School of Medical Sciences, University of Cape Coast, for their invaluable assistance, with special appreciation to Mr Philip Boakye Bonsu, Mr Albert Kwame Owusu, and Mr Ireneaus Nyame (Physiology Department).

## Author contributions

**Conceptualization:** Alice Charwudzi.

**Data curation:** Alice Charwudzi, Edward Morkporkpor Adela, Martin Ampofo, Aaron Fenuku, Abdul Raman Asemah, Edward Ahiakwah, Angela Animwaah Osei.

**Formal analysis:** Alice Charwudzi, Kingsley Kwadwo Asare Pereko, Abdul Raman Asemah.

**Investigation:** Edward Morkporkpor Adela, Daniel Edem Azumah.

**Supervision:** Emmanuel Kobina Mesi Edzie.

**Validation:** Alice Charwudzi, Aaron Fenuku, Emmanuel Kobina Mesi Edzie.

**Writing – original draft:** Alice Charwudzi, Abdul Raman Asemah, Angela Animwaah Osei.

**Writing – review & editing:** Alice Charwudzi, Edward Morkporkpor Adela, Kingsley Kwadwo Asare Pereko, Martin Ampofo, Aaron Fenuku, Daniel Edem Azumah, Edward Ahiakwah, Emmanuel Kobina Mesi Edzie.

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
