## [Decision Letter · Decision Letter 0]

21 May 2025

Dear Dr. Charwudzi,

Thank you for submitting your manuscript to PLOS ONE. After careful consideration, we feel that it has merit but does not fully meet PLOS ONE’s publication criteria as it currently stands. Therefore, we invite you to submit a revised version of the manuscript that addresses the points raised during the review process.

We look forward to receiving your revised manuscript.

Kind regards,

Enoch Aninagyei, PhD

Academic Editor

PLOS ONE

Journal Requirements:

2. We notice that your supplementary tables are included in the manuscript file. Please remove them and upload them with the file type 'Supporting Information'. Please ensure that each Supporting Information file has a legend listed in the manuscript after the references list.

Reviewers' comments:

Reviewer's Responses to Questions

**Comments to the Author**

1. Is the manuscript technically sound, and do the data support the conclusions?

Reviewer #1: Yes

Reviewer #2: Yes

Reviewer #3: Yes

2. Has the statistical analysis been performed appropriately and rigorously?

Reviewer #1: I Don't Know

Reviewer #2: Yes

Reviewer #3: Yes

3. Have the authors made all data underlying the findings in their manuscript fully available?

Reviewer #1: Yes

Reviewer #2: Yes

Reviewer #3: No

4. Is the manuscript presented in an intelligible fashion and written in standard English?

Reviewer #1: Yes

Reviewer #2: Yes

Reviewer #3: No

Reviewer #1: Hello.

I recently reviewed the manuscript “Prevalence and Predictors of Transfusion-Transmitted Infections Among Blood Donor Types at a Teaching Hospital in Ghana: Implications for Hemovigilance.” The work by Charwudzi et al. investigates the prevalence of transfusion-transmitted infections (TTIs) among various blood donor types. It critiques the reliance on rapid diagnostic tests (RDTs) to confirm infections in donated blood. While the article is well-written and presents engaging statistical analyses that captured my interest, it contains some shortcomings and errors, particularly in the discussion section, which I will address separately.

Abstract:

- The methods section in the abstract and the article's main text could benefit from additional information or examples of RDT tests to clarify the concept for readers.

- In the author summary section, it would be helpful for the author to explain concisely the rationale behind emphasizing NAT over RDTs.

Introduction:

- The text does not provide statistics on the prevalence of the mentioned viruses or TTIs within the geographical area of the study, which would enhance readers' understanding. Additionally, including a global prevalence statistic would offer a more comprehensive comparative perspective, especially since the text claims that the prevalence in the studied area is higher than the worldwide average.

- If prior studies have been conducted on the prevalence of TTIs in this geographical area or similar regions with comparable socio-economic conditions, briefly referencing those statistics would be beneficial. This allows for examining the trends in TTI prevalence and the global trend over time.

Materials and Methods:

- The section on ethical approval typically appears at the end of the document, preceding the references section.

- In line 126, the phrase "The study included data from 10,152 blood" should be revised to replace "blood" with "donated blood" or "blood sample."

- Could you clarify the meaning of "motivational incentives" or provide some examples?

- The parentheses at the end of line 156 were incorrectly placed and have been removed.

Results:

- In the Donor Demographics section, information for both donor groups should be consistent and uniform. This means that the percentage figures for either the student group or the age group should be included in lines 167 and 168. Alternatively, "3210" and "5188" should be removed from lines 169 and 170, respectively.

- In tables, the letter N, denoting the number, should be presented in uppercase.

- Explanations provided below the tables corresponding to the same table should be formatted in a smaller font to enhance clarity and distinguish them from the main text.

- In Table 4, the font for the numbers must be consistent throughout.

- In the Co-Infections section, no statistical analysis was performed to compare the two donor groups regarding the presence of a significant difference. What is the reason for this?

Discussion:

- Although the focus of the text, according to the explanations at the end of the Introduction section, is to examine the status of TTIs and their prevalence in different types of blood donors and to emphasize the weakness of RDTs for identifying infected donors (these issues are well addressed in the Discussion section), the main problem with this article is the Discussion section.

In the Results section, this study thoroughly examines HBV, HCV, HIV, and syphilis, with well-organized data presented in the tables. This approach leads the reader to anticipate speculation and discussion of these topics in the subsequent Discussion section. Even in some cases, the lack of significance of the difference between the two groups under study is worthy of discussion and speculation (for example, Table 2 of the HIV section in the Donation History section), but in the Discussion section, there is no mention of these data and no discussion is raised in this regard. It would be better to have a detailed discussion in this section regarding the data obtained from these infectious agents.

Images and diagrams:

- The quality of images 2 and 4 is currently low; it is advisable to provide improved versions. Additionally, the explanatory text overlaps with the image borders in certain sections of the flowchart or images, making it challenging to read. This issue should be addressed. Furthermore, for the text related to images 2 and 3, use the term "Figure" instead of "Fig" to ensure consistency across all images.

- The flowchart is also recommended to indicate the total number of subjects studied (10,152). At each stage, it would be beneficial to specify the number of subjects who tested positive for TTI and are no longer eligible for donation. This addition will enhance the reader's understanding of a significant aspect of the article's purpose and provide a more precise context before delving into the full text.

Reviewer #2: Greetings

nts

1- In this manuscript: you wrote ( Our)!!! The rule of manuscript

writing is to avoid using (Our). So you should delete (Our), use the formal

scientific words (This study or The current study or The present

study).

2- You wrote (We) several times!!! The rule of manuscript writing is to avoid

using (We). So you should delete (We), use the formal scientific words

(This study or The current study or The present

study).

Best regards.

Reviewer #3: Review for manuscript with “Title: Prevalence and predictors of transfusion-transmitted infections among blood donor types at a teaching hospital in Ghana: implications for hemovigilance”

Overall Assessment:

This is a well-structured and relevant study that explores the prevalence and predictors of transfusion-transmitted infections (TTIs) among different blood donor types in a teaching hospital in Ghana. The topic is extremely relevant, particularly when considering blood safety in environments with limited resources. The findings promote the adoption of more sensitive screening methods and the promotion of voluntary blood donation, both of which are important for public health. While the manuscript is generally strong in scope and presentation, a few areas need clarification and revision to meet PLOS ONE's standards, especially regarding methodological transparency, statistical presentation, and some grammatical and formatting improvements.

Strengths

1. The study addresses a significant public health issue in a resource-limited setting where transfusion safety is critical.

2. The sample size is large (15,683 donors), which enhances the statistical precision and reliability of the results.

3. A comprehensive picture of transfusion risk is provided by the analysis, which covers several TTI types (HIV, HBV, HCV, and syphilis).

4. By using multivariable logistic regression, the evaluation of predictors becomes strengthened.

5. The authors adequately explain the consent procedure and offer ethical approval.

Major issues:

Abstract

1. Donor categories lack clarity: "Voluntary walk-ins" and "voluntary mobiles" are two examples of terms that are not well defined. It could be defined more clearly for an international audience.

2. An incomplete explanation of the testing procedures. The number of donors examined using RDTs alone versus RDTs with ELISA, as well as the conditions under which ELISA was employed, are unknown. This makes the sensitivity gap (7.4%) reported in the results less interpretable.

3. Some statistical terms (e.g., adjusted odds ratios) are used without full context (confidence intervals are missing).

4. I recommend that the authors define all abbreviations at first use in the abstract—for example, "HBV (hepatitis B virus)," "HCV (hepatitis C virus)," and "RDTs (rapid diagnostic tests)"—to improve clarity for readers unfamiliar with these terms,

5. The mention that "ELISA confirmed an additional 7.4% of TTI cases among RDT non-reactive donors" is unclear as to additional detection by ELISA. Does this rate apply to RDT-negative donors, all tested donors, or 7.4% of all cases?

6. Although it is strongly advised, nucleic acid testing (NAT) is not contextualized. The financial, logistical, and technical viability of NAT in low-resource environments—all crucial for such a recommendation—is not discussed.

7. To improve the manuscript's accessibility and searching, I propose expanding the keywords by using more precise and standardized terminology such "ELISA," "repeat donors," and "risk factors."

Introduction:

1. The increasing incidence of transfusion-transmitted infections (TTIs) is discussed in the introduction, but it is unclear what exact requirement this study aims to fulfill, especially regarding the uniqueness or impact of comparing donor types in the Ghanaian circumstances.

2. Background information on Ghana's blood donation system, such as laws governing replacement vs. voluntary donations, current hemovigilance procedures, or the usage of screening technology, is lacking and is crucial for establishing the study's relevance.

3. Some sentences state well-known facts, such as that TTIs pose a threat to public health, without providing recent, pertinent citations or making a clear connection to the study issue.

4. Uncertain Rationale for Comparing RDT with ELISA:

The limitations of RDTs in practical blood screening, which would support the focus, are not sufficiently addressed in the justification for evaluating RDT performance in comparison to ELISA.

5. The Introduction does not conclude with a direct, clear summary of the study's goals or hypotheses to direct the reader to the Methods and Results.

Methods:

1. You wrote that ethical approval was changed in January 2025, which is in the future. However, your data gathering ended before that, so this is confusing. Check the dates to make sure they match the study's time frame.

2. You didn’t explain how you picked which donor records to include. For example, did you use all donors from 2022–2023? What about incomplete records? Clearly say who was included and why.

3. It’s not clear if repeat donors were counted more than once. This can make the infection rates look higher or lower than they are. Please, say whether the same person was counted once or every time they donated.

4. You state that ELISA was performed after non-reactive RDTs, which is not the standard procedure. The ELISA tests are most frequently used for positive outcomes; however, it is unclear whether you used the ELISA test for all donors or just a few.

5. You say you removed duplicates and flagged missing data, but you didn’t say how or how much was affected. Please, mention how many records were cleaned or what kinds of data problems you found.

6. Terms like “walk-in,” “mobile donor,” and “replacement donor” are used but not clearly defined. Briefly explain what each type means at the beginning (a small table could help).

7. You did not explain how you selected variables for logistic regression. Also, there is no mention of how you resolved missing data or whether you tested the reliability of the results. Include this information to help readers realize the reliability of your analysis.

8. Some donors received motivations (such as gifts), whereas others did not. This could affect who comes to give and influence the outcome. Mention this as a potential research limitation or bias. 9. There are minor grammatical errors and imprecise language. Proofread the section thoroughly to ensure it is clear and correct.

9. Please include brand information, sensitivity and specificity, and internal quality control techniques for ELISA testing. Additionally, a List of manufacturers' indicators of performance for RDT kits. This is crucial, given the significant difference between RDT and ELISA results.

Result:

1. The text contains many of the exact numbers as the tables. Please, only highlight the most essential findings in the text. Allow the tables to reveal the details.

2. The results are disorganized, jumping from one topic to another. Use subheadings such as "Donor Characteristics," "TTI Prevalence," "Co-infections,"...etc. to guide the reader.

3. The number of co-infections (those who have more than one infection) is uncertain. It's unclear whether any contributors were included twice. Explain exactly how these numbers were counted and whether they overlap.

4. In certain areas, the total number of contributors is 10,152, whereas in others it is 8,356. There is no explanation. Explain why some data is missing (for example, not all donors provided blood group information).

5. Some figures are mentioned in the text (such as Figures 2, 3, and 4), but they are not shown or explained in detail. Ensure that all figures are included and clearly defined in the text.

6. P-values (e.g., p < 0.001) are provided; however, crucial statistics such as odds ratios and confidence intervals are either missing or not explained. Please, include these numbers and explain them in simple terms.

7. The tables lack clarity in style, and several headers are unclear. Apply the same structure to all tables and clarify all terminology and abbreviations.

8. Although the logistic regression model (Table 4) is presented, the authors do not provide an explanation for their actions. Give a brief explanation of the model's elements and testing methods.

9. Depending on the context, the word "TTI" can refer to any infection or a group of infections. I suggested that the term "overall TTI" be defined precisely and consistently.

10. The frequency of each infection is reported by the authors; however, they exclude confidence intervals, which indicate how accurate the figures are. I suggested that 95% confidence intervals be included with each percentage.

11. Group differences (students versus employees, for example) are explained without addressing further potential reasons.

Recommendation: When describing these patterns, exercise greater caution and address relevant confounding variables.

12. The Results section should just focus on the findings. However, in some cases, the authors interpret the meaning. Remove any opinions or interpretations to the Discussion section.

13. Words such as "prevalence" and "proportion" are used in ways that may be unclear to readers. Use "prevalence" when discussing how widespread a condition is in the population and "proportion" when referring to specific areas of the study sample.

14. Many p-values are displayed, although it is unclear whether the authors corrected for conducting numerous tests. Explain whether they adjusted for repeated comparisons to prevent false findings.

15. There are various grammatical errors and confusing sentences. Suggestion: Carefully revise the text to improve clarity and professionalism.

Discussion:

1. The discussion part requires further integration and comparison with findings from local and international studies. Currently, the discussion repeats findings without properly contextualizing them within the larger literature on TTIs in Sub-Saharan Africa and related regions.

Please check your prevalence data (particularly the greater syphilis incidence) against national averages or WHO estimates and investigate possible explanations for the discrepancy.

2. The consequences of high syphilis prevalence are understudied. Are these rates consistent with Ghana's population-level STI trends? Could there be screening concerns (for example, false positives from RDTs).

The discussion of the limitations of rapid diagnostic tests (RDTs) is highlighted but needs to be expanded. You discovered that 7.4% of TTIs were missed by RDTs but later validated by ELISA; this fact requires a more rigorous interpretation and consideration in terms of screening policy.

3. In some cases, the results are merely restated without further analysis. Talking about statistics isn't enough; we also need to look at the reasons behind these trends, particularly about donor types and sociodemographic.

Conclusion

a. The conclusion includes broad recommendations (e.g., adoption of NAT and policy adjustments to voluntary donation) without taking into consideration feasibility restrictions in low-resource environments.

While NAT is ideal, the report should clearly state the actual challenges to its implementation in Ghana (cost, infrastructure, and training).

b. The statement that the findings had "significant implications for hemovigilance" is unclear. Please describe what changes should be made (e.g., prioritizing voluntary donors, discontinuing RDTs, improving donor education, etc.).

Ethical Considerations

There are a few ethical issues that need to be addressed, including a discrepancy in the ethical approval dates, little information on how donor data was anonymized and protected, and an insufficient explanation for the withdrawal of informed consent.

Minor Issues:

1. The title is clear and informative. Few changes are necessary to comply with PLOS ONE formatting rules, such as uniform formatting of author names, proper use of superscript numbers for affiliations, and limiting the corresponding author to one professional email address.

2. Several typographical and grammatical problems can be detected throughout the work; rigorous editing is recommended.

3. Define all abbreviations at the beginning (e.g., NAT, ELISA).

4. Inconsistencies in statistical reporting (such as p-values) and reference formatting should be addressed.

5. Figures and tables require better formatting, labeling, and the addition of missing information. 6. Language usage should be standardized (for example, distinguishing between British and American English).

7. The approval timeline looks to be uncertain, anonymization techniques require greater explanation, and the justification for waiving informed consent should be strengthened and related to ethical standards.

Recommendation: Major revision.

**Do you want your identity to be public for this peer review?** For information about this choice, including consent withdrawal, please see our Privacy Policy

Reviewer #1: No

Reviewer #2: No

Reviewer #3: No

---

## [Author Response · Author response to Decision Letter 1]

3 Jul 2025

REVIEWER #1

Comment: While the article is well-written and presents engaging statistical analyses that captured my interest, it contains some shortcomings and errors, particularly in the discussion section, which I will address separately.

Response: We thank the reviewer for the constructive comment.

Abstract:

Comment: The methods section in the abstract and the article's main text could benefit from additional information or examples of RDT tests to clarify the concept for readers.

Response: We are grateful to the esteemed reviewer for this comment. This additional information about the RDTs used (e.g., Determine™ for HIV, SD Bioline™ for HBsAg and HCV, and yphilis Rapid Test Strip™) has been provided.

Comment: In the author summary section, it would be helpful for the author to explain concisely the rationale behind emphasising NAT over RDTs.

Response: We expanded the author summary to briefly explain that NAT offers improved sensitivity, particularly during the early window period of infection, but remains cost-prohibitive in low-resource settings. Thus, ELISA is presented as a more feasible alternative for improving blood safety.

Introduction:

Comment: The text does not provide statistics on the prevalence of the mentioned viruses or TTIs within the geographical area of the study, which would enhance readers' understanding. Additionally, including a global prevalence statistic would offer a more comprehensive comparative perspective, especially since the text claims that the prevalence in the studied area is higher than the worldwide average.

Response: Thank you for these comments. The entire introduction section has been extensively modified to reflect all the concerns raised, in addition to provided more information to make it more comprehensible.

Comment: If prior studies have been conducted on the prevalence of TTIs in this geographical area or similar regions with comparable socio-economic conditions, briefly referencing those statistics would be beneficial. This allows for examining the trends in TTI prevalence and the global trend over time.

Response: We are grateful for these comments. More information with regards to this topic in our geographical area or similar regions has been included and referenced accordingly, including Tetteh et al. (2018) and Okoroiwu et al. (2021). This enhances the reader’s understanding of the local context.

Materials and Methods:

Comment: The section on ethical approval typically appears at the end of the document, preceding the references section..

Response: Thank you for the observation. As per journal format, the ethical issues or considerations have been moved to the end of the methodology section in agreement with journal guidelines.

Comment: In line 126, the phrase "The study included data from 10,152 blood" should be revised to replace "blood" with "donated blood" or "blood sample."

Response: The entire methodology section has been extensively modified to reflect all the concerns raised. The sentence now reads: “The study included data from 10,152 potential blood donors…

Comment: Could you clarify the meaning of "motivational incentives" or provide some examples?

Response: We now clarify that “motivational incentives” refer to post-donation refreshments and small souvenir items (e.g., exercise books) provided to voluntary donors to encourage participation.

Comment: The parentheses at the end of line 156 were incorrectly placed and have been removed

Response: The parentheses at the end of line 156 have been removed appropriately.

Results:

Comment: In the Donor Demographics section, information for both donor groups should be consistent and uniform. This means that the percentage figures for either the student group or the age group should be included in lines 167 and 168. Alternatively, "3210" and "5188" should be removed from lines 169 and 170, respectively

Response: Consistency has been ensured in the presentation of the results. All the necessary modifications have been made.

Comment: In tables, the letter N, denoting the number, should be presented in uppercase.

Response: All instances of “n” have been revised to uppercase “N” in all the tables.

Comment: Explanations provided below the tables corresponding to the same table should be formatted in a smaller font to enhance clarity and distinguish them from the main text.

Response: We are grateful for the comment. We reformatted all table footnotes according to the journal’s guidelines.

Comment: In Table 4, the font for the numbers must be consistent throughout.

Response: The font in Table 3 (previously Table 4) has been reviewed and standardised to ensure consistency across all numeric entries, and the font size has been maintained at 11 for consistency.

Comment: In the Co-Infections section, no statistical analysis was performed to compare the two donor groups regarding the presence of a significant difference. What is the reason for this?

Response: The necessary associations to compare the two donor groups have been made and added to the write-up at the co-infections section. The statistical comparison of co-infection rates between donor groups yielded a non-significant result (p=0.648).

Discussion:

Comment: Although the focus of the text, according to the explanations at the end of the Introduction section, is to examine the status of TTIs and their prevalence in different types of blood donors and to emphasize the weakness of RDTs for identifying infected donors (these issues are well addressed in the Discussion section), the main problem with this article is the Discussion section.

Response: We appreciate the reviewer for these comments. From the comments it is a bit confusing what the esteemed review wanted to convey but we assumed that it was relating to the improvement of the discussion relating to TTIs and their prevalence in different types of blood donors, which have been duly improved.

Comment: In the Results section, this study thoroughly examines HBV, HCV, HIV, and syphilis, with well-organized data presented in the tables. This approach leads the reader to anticipate speculation and discussion of these topics in the subsequent Discussion section. Even in some cases, the lack of significance of the difference between the two groups under study is worthy of discussion and speculation (for example, Table 2 of the HIV section in the Donation History section), but in the Discussion section, there is no mention of these data and no discussion is raised in this regard. It would be better to have a detailed discussion in this section regarding the data obtained from these infectious agents.

Response: We agree with the reviewer and have expanded the discussion to interpret the findings for each TTI, syphilis, HBV, HCV, and HIV, including cases where no statistically significant difference was observed, under the heading “The prevalence of syphilis, HBV, HCV and HIV” to improve the discussion section.

Images and diagrams:

Comment: The quality of images 2 and 4 is currently low; it is advisable to provide improved versions. Additionally, the explanatory text overlaps with the image borders in certain sections of the flowchart or images, making it challenging to read. This issue should be addressed. Furthermore, for the text related to images 2 and 3, use the term "Figure" instead of "Fig" to ensure consistency across all images.

Response: We have improved the image quality for Figures 2 and 4 with high-resolution versions, and the image quality meets the journal requirement when checked on NAAS (https://ngplosjournals.pagemajik.ai/artanalysis). We have also improved spacing in the flowchart to eliminate text overlap, and standardised terminology throughout the manuscript to use “Fig” as per journal style.

Comment: The flowchart is also recommended to indicate the total number of subjects studied (10,152). At each stage, it would be beneficial to specify the number of subjects who tested positive for TTI and are no longer eligible for donation. This addition will enhance the reader's understanding of a significant aspect of the article's purpose and provide a more precise context before delving into the full text.

Response: The revised flowchart now begins with the full donor cohort (N = 10309) and details the number of TTI-positive donors excluded at each stage. This enhances transparency and guides the reader through the donor inclusion process.

REVIEWER #2

Comment 1- In this manuscript: you wrote ( Our)!!! The rule of manuscript writing is to avoid using (Our). So you should delete (Our), use the formal scientific words (This study or The current study or The present study).

Response:

Thank you for your observation. We have revised the manuscript thoroughly and replaced all instances of the term “our” with more appropriate and formal alternatives such as “this study” or “the current study,” as recommended.

Comment 2: You wrote (We) several times!!! The rule of manuscript writing is to avoid using (We). So you should delete (We), use the formal scientific words (This study or The current study or The present study).

Response: We appreciate this suggestion. We have carefully revised the manuscript and replaced all occurrences of “we” with formal, third-person expressions, such as “the study” or “the current study,” per standard scientific writing conventions.

REVIEWER #3

Comment: This is a well-structured and relevant study that explores the prevalence and predictors of transfusion-transmitted infections (TTIs) among different blood donor types in a teaching hospital in Ghana. The topic is extremely relevant, particularly when considering blood safety in environments with limited resources. The findings promote the adoption of more sensitive screening methods and the promotion of voluntary blood donation, both of which are important for public health. While the manuscript is generally strong in scope and presentation, a few areas need clarification and revision to meet PLOS ONE's standards, especially regarding methodological transparency, statistical presentation, and some grammatical and formatting improvements.

Response: We appreciate the reviewer for the positive feedback on our manuscript and the constructive suggestions to help improve the manuscript.

Comment: The study addresses a significant public health issue in a resource-limited setting where transfusion safety is critical. 2. The sample size is large (15,683 donors), which enhances the statistical precision and reliability of the results. 3. A comprehensive picture of transfusion risk is provided by the analysis, which covers several TTI types (HIV, HBV, HCV, and syphilis). 4. By using multivariable logistic regression, the evaluation of predictors becomes strengthened. 5. The authors adequately explain the consent procedure and offer ethical approval.

Responses: We appreciate the reviewer for the positive feedback on our manuscript, which is very encouraging.

Major issues:

Abstract

Comment 1:

Comment 1: Donor categories lack clarity: "Voluntary walk-ins" and "voluntary mobiles" are two examples of terms that are not well defined. It could be defined more clearly for an international audience.

Response: The methodology section has been extensively revised to provide more clarity on the donor categories. Specifically, “voluntary walk-ins” refer to individuals who donate at the fixed-site blood bank without external mobilisation, while “voluntary mobile donors” are individuals recruited through outreach drives at schools, churches, or public locations.

Comment 2: An incomplete explanation of the testing procedures. The number of donors examined using RDTs alone versus RDTs with ELISA, as well as the conditions under which ELISA was employed, are unknown. This makes the sensitivity gap (7.4%) reported in the results less interpretable.

Response: We are grateful for this comment. The methodology section has been extensively revised to provide more context on the proportion of donors tested with RDT only versus RDT and ELISA for clarity. More details have also been added to the flow chart and supplementary Table 1 (S1 Table) to aid with the clarification.

Comment 3: Some statistical terms (e.g., adjusted odds ratios) are used without full context (confidence intervals are missing).

Response: 95% confidence intervals have now been added to the adjusted odds ratios.

Comment 4: I recommend that the authors define all abbreviations at first use in the abstract—for example, "HBV (hepatitis B virus)," "HCV (hepatitis C virus)," and "RDTs (rapid diagnostic tests)"—to improve clarity for readers unfamiliar with these terms,

Response: Done. All abbreviations used in the abstract and throughout the manuscript are now defined at first mention.

Comment 5: The mention that "ELISA confirmed an additional 7.4% of TTI cases among RDT non-reactive donors" is unclear as to additional detection by ELISA. Does this rate apply to RDT-negative donors, all tested donors, or 7.4% of all cases?

Response: We are grateful for the comment. This is an additional detection percentage, which is actually 7.3% (calculation error), obtained after retesting of non-reactive (negative) RDT samples using ELISA. The explanation has been added to the write-up under “Performance for screening methods” of the results section and the flow chart.

Comment 6: Although it is strongly advised, nucleic acid testing (NAT) is not contextualized. The financial, logistical, and technical viability of NAT in low-resource environments—all crucial for such a recommendation—is not discussed.

Response: We have added a sentence to the Author Summary and Discussion noting that while NAT is ideal, its use is constrained in low-resource settings like Ghana due to financial and infrastructural limitations. ELISA is thus presented as a more feasible alternative.

Comment 7: To improve the manuscript's accessibility and searching, I propose expanding the keywords by using more precise and standardized terminology such "ELISA," "repeat donors," and "risk factors."

Response: Keywords have been modified to improve discoverability and comply with Medical Subject Headings (MESH).

Introduction

Comment 1: The increasing incidence of transfusion-transmitted infections (TTIs) is discussed in the introduction, but it is unclear what exact requirement this study aims to fulfill, especially regarding the uniqueness or impact of comparing donor types in the Ghanaian circumstances.

Response: We revised the Introduction to state that the study aims to fill a gap in understanding the differential risk of TTIs among donor types (voluntary vs. replacement) in the context of Ghana, where replacement donation is still widely practised. The rationale for comparing these groups is now made explicit.

Comment 2: Background information on Ghana's blood donation system, such as laws governing replacement vs. voluntary donations, current hemovigilance procedures, or the usage of screening technology, is lacking and is crucial for establishing the study's relevance.

Response: We added a paragraph summarising Ghana’s current blood donation policies, noting that:

• Ghana’s blood donation system is largely governed by the National Blood Service Act,

• Both voluntary and replacement donors are used due to supply constraints.

• Haemovigilance efforts are evolving, but a national donor deferral registry and unified data system are lacking.

• RDTs remain the mainstay of screening in most centres, with limited use of ELISA or NAT due to resource constraints.

Comment 3: Some sentences state well-known facts, such as that TTIs pose a threat to public health, without providing recent, pertinent citations or making a clear connection to the study issue.

Response: The entire manuscript has been thoroughly perused, and significant statements made have been duly cited and referenced.

Comment 4: Uncertain Rationale for Comparing RDT with ELISA: The limitations of RDTs in practical blood screening, which would support the focus, are not sufficiently addressed in the justification for evaluating RDT performance in comparison to ELISA.

Response: We are grateful for this comment. We wanted to find the performance effectiveness of RDT usag

---

## [Decision Letter · Decision Letter 1]

8 Aug 2025

Dear Dr. Alice Charwudzi,

Thank you for submitting your manuscript to PLOS ONE. After careful consideration, we feel that it has merit but does not fully meet PLOS ONE’s publication criteria as it currently stands. Therefore, we invite you to submit a revised version of the manuscript that addresses the points raised during the review process.

We look forward to receiving your revised manuscript.

Kind regards,

Enoch Aninagyei, PhD

Academic Editor

PLOS ONE

Journal Requirements:

Reviewers' comments:

Reviewer's Responses to Questions

**Comments to the Author**

Reviewer #1: All comments have been addressed

2. Is the manuscript technically sound, and do the data support the conclusions?

Reviewer #1: Yes

3. Has the statistical analysis been performed appropriately and rigorously?

Reviewer #1: N/A

4. Have the authors made all data underlying the findings in their manuscript fully available?

Reviewer #1: Yes

5. Is the manuscript presented in an intelligible fashion and written in standard English?

Reviewer #1: Yes

Reviewer #1: The revised article is deemed satisfactory and acceptable. The modifications implemented have significantly enhanced the overall quality of the study and elevated its scientific rigor. All peer-review concerns have been addressed and revised appropriately. The sole remaining minor issue is the lowercase representation of "n" (denoting number) in Table 4, which should be consistent with the journal's writing guidelines and, if necessary, capitalized.

**Do you want your identity to be public for this peer review?** For information about this choice, including consent withdrawal, please see our Privacy Policy

Reviewer #1: No

---

## [Author Response · Author response to Decision Letter 2]

19 Aug 2025

Editor-in-Chief

PLOS ONE

15th August 2025

Dear Editor,

Re: Revised manuscript submission manuscript number: PONE-D-25-18662R1

Title: Prevalence and predictors of transfusion-transmitted infections among blood donor types at a teaching hospital in Ghana: implications for haemovigilance.

We are sincerely grateful to you and the Reviewers for your time, insightful comments, and constructive feedback, which have greatly improved the quality and rigour of our manuscript. We also thank you for the opportunity to revise and resubmit our work to PLOS ONE.

The manuscript has been carefully revised to address all comments, and our responses are provided point by point. We hope that the revised version meets the requirements for publication in PLOS ONE, and appreciate your consideration of our resubmission.

Yours sincerely,

Dr Alice Charwudzi

Corresponding author

Reviewer comments and author responses

Journal Requirements:

Comment 1: If the reviewer comments include a recommendation to cite specific previously published works, please review and evaluate these publications to determine whether they are relevant and should be cited. There is no requirement to cite these works unless the editor has indicated otherwise.

Response: We did not receive any recommendations from the reviewers or the editor to cite any specific previously published articles.

Comment 2: Please review your reference list to ensure that it is complete and correct. If you have cited papers that have been retracted, please include the rationale for doing so in the manuscript text, or remove these references and replace them with relevant current references. Any changes to the reference list should be mentioned in the rebuttal letter that accompanies your revised manuscript. If you need to cite a retracted article, indicate the article’s retracted status in the References list and also include a citation and full reference for the retraction notice.

Response: We carefully reviewed the reference list to ensure accuracy and completeness. During revision, some references were reshuffled, citations that were redundant or did not fully support specific statements were removed from those sentences, but retained elsewhere in the manuscript where they remain relevant. All references were verified against PubMed, the respective journal websites, and the Retraction Watch Database (accessed 12 August 2025). None of the cited works had been retracted as of this date.

REVIEWER #1

Comment: The revised article is deemed satisfactory and acceptable. The modifications implemented have significantly enhanced the overall quality of the study and elevated its scientific rigor. All peer-review concerns have been addressed and revised appropriately. The sole remaining minor issue is the lowercase representation of "n" (denoting number) in Table 4, which should be consistent with the journal's writing guidelines and, if necessary, capitalized.

Response: We thank the reviewer for this observation. The “n” has been capitalised in Table 4 per the journal’s style.

Additional Corrections

Figure 1 Legend: Revised slightly for clarity. An asterisk was added, and the legend now reads:

Flowchart of the recruitment and screening process for 10,152 potential blood donors at a teaching hospital in the Central Region of Ghana. *Donors with incomplete data at initial extraction were excluded without record retention. Of 10,309 captured records, 159 (1.5%) with incomplete fields were removed, leaving 10,152 for analysis.

Table 4 Correction: On re-checking, we identified a transcription error in the “Prev N (%)” column. The numerators (N) were incorrectly copied, although the percentages and logistic regression outputs (OR, 95% CI, p-values) were based on the correct dataset. The corrected values now align with the denominator (N = 4,451) and percentages. This correction does not affect statistical significance or study conclusions.

Proofreading: We extensively proofread the manuscript, and necessary corrections have been made and tracked.

We believe that the revisions have fully addressed all reviewer and editorial concerns. We thank you again for your consideration.

Thank you.

---

## [Editor Report · Decision Letter 2]

1 Sep 2025

Dear Dr. Alice Charwudzi,

Thank you for submitting your manuscript to PLOS ONE. After careful consideration, we feel that it has merit but does not fully meet PLOS ONE’s publication criteria as it currently stands. Therefore, we invite you to submit a revised version of the manuscript that addresses the points raised during the review process.

**Background**

Provide baseline data on Ghana’s TTI prevalence from prior studies (e.g., HBV, HCV, HIV, syphilis rates) to situate your findings in context.Include mention of the *window period risk* (missed infections due to early-stage infection) as a limitation of RDTs and ELISAs compared to NAT.Add regional context — e.g., "Previous studies in Ghana report TTI prevalence ranging from X–Y%, but reliance on RDTs may underestimate burden."

**Objective**

Refine objective wording for precision: Instead of "assessed the prevalence and predictors of TTIs," specify that you **estimated prevalence, identified sociodemographic and donor-type predictors, and compared diagnostic yield of RDTs vs ELISA** .

**Methods**

Clarify the **confirmatory testing strategy** : Only RDT-non-reactive samples underwent ELISA. Were RDT-reactive samples confirmed with ELISA? If not, prevalence estimates may be inflated due to false positives.Address potential **bias between fixed-site and mobile donors** : Since the two groups used different screening strategies (RDT+ELISA vs ELISA-only), diagnostic sensitivity may differ, affecting prevalence comparisons.Specify all **covariates included in multivariable regression** (e.g., age, sex, education, occupation, donor type, donation frequency) for transparency and reproducibility.Define **repeat donation criteria** clearly (≥2 donations in study period vs. documented donor history).

**Results/Interpretation**

Emphasize that **donor type was not a significant predictor** after adjusting for sociodemographic factors, highlighting confounding effects.Discuss **RDT false positives** : Without confirmatory ELISA for reactive cases, diagnostic accuracy analysis is incomplete.Explore whether **fixed-site vs mobile donors differ demographically** (age, sex, etc.), since these differences could confound prevalence estimates.

**Discussion/Conclusion**

Add **cost and logistical implications** of switching from RDTs to ELISA in Ghana, as policymakers will require feasibility considerations.Mention **NAT (nucleic acid testing)** as the gold standard in blood safety. Even if not currently available in Ghana, acknowledging it strengthens the discussion on diagnostic hierarchy.

We look forward to receiving your revised manuscript.

Kind regards,

Enoch Aninagyei, PhD

Academic Editor

PLOS ONE
---

## [Author Response · Author response to Decision Letter 3]

2 Oct 2025

Editor-in-Chief

PLoS ONE

15th September 2025

Dear Editor,

Re: Revised manuscript submission manuscript number: PONE-D-25-18662R2

Title: Prevalence and predictors of transfusion-transmitted infections among blood donor types at a teaching hospital in Ghana: implications for haemovigilance.

We thank the Editor and Reviewers for their constructive feedback, which has helped us strengthen the manuscript. Below, we provide a point-by-point response to each comment, indicating how the manuscript was revised.

We hope that the revised version meets the requirements for publication in PLoS ONE, and appreciate your consideration of our resubmission.

Yours sincerely,

Dr Alice Charwudzi

Corresponding author

Reviewer comments and author responses

Background

Comment 1: Provide baseline data on Ghana’s TTI prevalence from prior studies (e.g., HBV, HCV, HIV, syphilis rates) to situate your findings in context.

Response: We expanded the Introduction (lines 77-86) to include prevalence estimates from multiple Ghanaian studies. Studies were selected for methodological robustness and similarity in the methodology in terms of the examined pathogens. Although Hamidu et al. (2023) (https://doi.org/10.33140/IJHPP.02.02.05) and Walana et al. (2014) (https://doi.org/10.9734/BMRJ/2014/12160) reported on the prevalence of TTIs in Ghana, they assessed only the viral pathogens, while this present study considered all four major TTIs; thus, they were excluded.

Comment 2: Include mention of the window period risk (missed infections due to early-stage infection) as a limitation of RDTs and ELISAs compared to NAT.

Response: We revised the Introduction (lines 97-104) to highlight that both RDTs and ELISA may miss early infections during the window period, whereas NAT offers superior sensitivity.

Comment 3: Add regional context — e.g., "Previous studies in Ghana report TTI prevalence ranging from X–Y%, but reliance on RDTs may underestimate burden."

Response: This has been addressed in the Introduction (lines 77-86 and lines 97-103), with emphasis on regional prevalence and limitations of RDT-based screening.

Objective

Comment 4: Refine objective wording for precision: Instead of "assessed the prevalence and predictors of TTIs," specify that you estimated prevalence, identified sociodemographic and donor-type predictors, and compared diagnostic yield of RDTs vs ELISA.

Response: The Abstract (lines 29-31) and Introduction (lines 105-109) now specify that we estimated prevalence, identified sociodemographic and donor-type predictors, and compared the diagnostic yield of RDTs versus ELISA.

Methods

Comment 5: Clarify the confirmatory testing strategy: Only RDT-non-reactive samples underwent ELISA. Were RDT-reactive samples confirmed with ELISA? If not, prevalence estimates may be inflated due to false positives.

Response: We clarified that RDT-reactive samples were not confirmed with ELISA, per national protocol. This is stated in the Method (lines 172-176), reiterated in the Discussion (lines 431-434), and noted in the Limitations (lines 530-537).

Comment 6: Address potential bias between fixed-site and mobile donors: Since the two groups used different screening strategies (RDT+ELISA vs ELISA-only), diagnostic sensitivity may differ, affecting prevalence comparisons.

Response: We have now rephrased the portions that acknowledged the differing screening strategies may bias prevalence comparisons (lines 179-181), and rephrased the limitation (lines 533-537).

Comment 7: Specify all covariates included in multivariable regression (e.g., age, sex, education, occupation, donor type, donation frequency) for transparency and reproducibility.

Response: We added the list of covariates (age, gender, donation frequency, donor type and occupation) to the Statistical Analysis section (lines 217-218).

Comment 8: Define repeat donation criteria clearly (≥2 donations in study period vs. documented donor history).

Response: We clarified that repeat donors were defined as individuals with ≥1 prior donation history before the study period (Methods, lines 136-137).

Results/interpretation

Comment 9: Emphasize that donor type was not a significant predictor after adjusting for sociodemographic factors, highlighting confounding effects.

Response: This is now emphasised in Results (lines 344-346) and Discussion (lines 466-468).

Comment 10: Discuss RDT false positives: Without confirmatory ELISA for reactive cases, diagnostic accuracy analysis is incomplete.

Response: Your concerns were already included in the original manuscript, but we have further done slight rephrasing and introduced additional information in the Methods (lines 172-176), Discussion (lines 431-434), and Limitations (lines 530-537), noting that the lack of an ELISA confirmation for RDT-reactive samples may inflate prevalence estimates.

Comment 11: Explore whether fixed-site vs mobile donors differ demographically (age, sex, etc.), since these differences could confound prevalence estimates.

Response: We compared the two donor groups and noted key demographic differences in the Results (lines 253-256 and lines 301-304), and Discussion (lines 470-472). Supporting data are provided in the Supplementary Table (S1B Table).

Discussion/Conclusion

Comment 12: Add cost and logistical implications of switching from RDTs to ELISA in Ghana, as policymakers will require feasibility considerations.

Response: We expanded the discussion (lines 500-512) to include cost estimates and logistical considerations for ELISA implementation.

Comment 13: Mention NAT (nucleic acid testing) as the gold standard in blood safety. Even if not currently available in Ghana, acknowledging it strengthens the discussion on diagnostic hierarchy.

Response: NAT is now acknowledged as the most sensitive method for blood safety in the Introduction (lines 97-104), Discussion (lines 505-512), Conclusion (lines 566-567) and the Abstract’s conclusion (lines 49-52)

Additional corrections

Some reviewer concerns had already been addressed in the earlier version. To avoid ambiguity, we have rephrased relevant sections to more explicitly align with the feedback. The manuscript has been extensively proofread to remove redundancies and improve clarity. All changes are tracked for transparency.

Thank you.

---

## [Editor Report · Decision Letter 3]

13 Oct 2025

Prevalence and predictors of transfusion-transmitted infections among blood donor types at a teaching hospital in Ghana: implications for haemovigilance.

PONE-D-25-18662R3

Dear Dr. Alice Charwudzi,

We’re pleased to inform you that your manuscript has been judged scientifically suitable for publication and will be formally accepted for publication once it meets all outstanding technical requirements.

Kind regards,

Enoch Aninagyei, PhD

Academic Editor

PLOS ONE
---

## [Editor Report · Acceptance letter]

PONE-D-25-18662R3

PLOS ONE

Dear Dr. Charwudzi,

I'm pleased to inform you that your manuscript has been deemed suitable for publication in PLOS ONE. Congratulations! Your manuscript is now being handed over to our production team.

Kind regards,

on behalf of

Dr Enoch Aninagyei

Academic Editor

PLOS ONE